# Effects of Nutritional Disturbances on the Structure and Function of Mitochondria, Oxidative Stress Level, and Fat Deposition in Chicken Liver Cells

**DOI:** 10.3390/ani15213151

**Published:** 2025-10-30

**Authors:** Suyan Zhu, Pei Zhang, Ya Xing, Xiaoyi Zhou, Jing Ge, Xiaoxu Jia, Yushi Gao, Tuoyu Geng

**Affiliations:** 1College of Animal Science and Technology, Yangzhou University, Yangzhou 225009, China; zsy2738120083@163.com (S.Z.); peizhang202505@163.com (P.Z.); xingya325@163.com (Y.X.); 13218820235@163.com (X.Z.); gejing@yzu.edu.cn (J.G.); 2Jiangsu Institute of Poultry Sciences, Yangzhou 225100, China; 2023405004@stu.njau.edu.cn; 3Joint International Research Laboratory of Agriculture and Agri-Product Safety of the Ministry of Education of China, Yangzhou 225009, China

**Keywords:** chicken, nutrition, mitochondrial haplogroup, maternal effect, oxidative stress

## Abstract

**Simple Summary:**

Nutrition and energy are critical factors influencing animal growth, development, production performance, and health status; however, the understanding of the underlying mechanism remains limited. In this study, we found that refeeding after fasting promoted fat deposition and increased mitochondrial membrane potential and the number of clustered ribosomes in liver cells. However, mitochondrial quantity and cellular ROS level were influenced by different mitochondrial haplogroups (A and E Haplogroups). Moreover, Haplogroup A chicken liver cells exhibited stronger adaptability to nutritional disturbances than Haplogroup E. In addition, glucose and oleic acid treatments exerted distinct effects on mitochondria membrane potential, ROS level and CYTB protein abundance in liver cells. These findings provide an insight into the mechanism by which nutritional disturbances influence cell physiological functions and, ultimately, animal performance and health.

**Abstract:**

As mitochondria play an important role in nutritional/energy metabolism, nutritional disturbances may affect animal growth, development and performance through modulating mitochondrial structure and function. This study aimed to elucidate the effects of nutritional disturbances on mitochondrial structure and function, oxidative stress, and fat deposition in the hepatocytes of chickens with A or E mitochondrial haplogroups (referred to as A-group and E-group). For in vivo experiments, white-feathered broiler chickens were fasted for 12 h or refed for 2 h after 10 h fasting. For in vitro experiments, chicken embryonic primary hepatocytes were treated with 50 mmol/L glucose or 0.25 mmol/L oleic acid. Data indicated that compared to fasted chickens, fat content (*p* < 0.01), the number of aggregated ribosomes (*p* < 0.05), and mitochondrial membrane potential (*p* < 0.05) were increased in the refed chickens of both haplogroups. However, the number of mitochondria was reduced (*p* < 0.01) and ROS level was increased (*p* < 0.05) in the refed E-group chickens, and the protein levels of MFN2 and SOD2 were reduced (*p* < 0.05) in the refed A-group chickens. Moreover, compared to the control cells, triglyceride content was increased in the cells of both haplogroups (*p* < 0.01), ROS level was reduced in the E-group cells (*p* < 0.01), and mitochondrial membrane potential was reduced (*p* < 0.05) and CYTB protein content was increased (*p* < 0.05) in the A-group cells after treatment with oleic acid. In addition, mitochondrial membrane potential was increased in the A-group cells after treatment with glucose (*p* < 0.01). These results indicate that nutritional disturbances affected fat deposition, mitochondrial membrane potential, the number of aggregated ribosomes, and ROS level in chicken liver cells. Moreover, ROS level, mitochondrial number, mitochondrial membrane potential, and the abundance of certain mitochondrial proteins were different between the A- and E-groups or between glucose and oleic acid treatments. These findings provide references for improving animal physiological functions and production performance by adjusting nutritional levels.

## 1. Introduction

The egg and meat products of poultry constitute a primary source of high-quality protein for humans; therefore, the expansion of the poultry sector is critical for addressing worldwide nutritional deficiencies. Currently, the scale of the poultry industry is immense, with total chicken meat yield reaching 26.37 million tons in 2024, statistics released by the Ministry of Agriculture and Rural Affairs of China (based on the monthly fixed-point tracking monitoring data on broiler farmers and the monitoring data from the China Animal Industry Association). Facing such a large-scale poultry industry, how to further improve the production performance of poultry is an important issue.

Nutrition and energy are pivotal factors influencing animal growth and development, productivity and health status. Chronic undernutrition may cause growth retardation, diminished productivity and compromised stress/disease resistance. Conversely, prolonged overnutrition can lead to obesity, metabolic disorders, reduced feed utilization and reproductive performance, etc. Therefore, adjusting nutrition and energy appropriately according to the different growth and development stages, production phases, and health conditions of animals may help promote their growth and development, boost production yields and feed efficiency, and ameliorate health outcomes. Nevertheless, understanding of the mechanisms by which nutrition and energy regulate animal growth, productivity, and health remains limited. Revealing the underlying mechanisms may provide useful information for adjusting nutrition and energy to maximize animal production efficiency.

A variety of animal models are employed in studies investigating nutritional/energy metabolism. The fasting/refeeding model has been widely used in elucidating how nutritional disturbances affect animal growth, productivity and health status. A previous study has demonstrated that a 2 h fasting significantly suppresses hepatic lipogenesis in chicks, and this inhibitory effect is reversed after refeeding. Similarly, high-fat/high-protein diets can also suppress lipogenesis [1]. Moreover, Weng et al. observed that fasting markedly decreased body weight, liver weight and hepatosomatic index (liver-to-body weight ratio) in laying hens, while refeeding could prevent this decrease [2]. In addition, these models are also employed to elucidate the mechanisms by which animals adapt to nutritional disturbances during scenarios such as starvation caused by restricted feeding or long-distance transport, as well as overfeeding.

The liver plays a pivotal role in animal energy and substance metabolisms, orchestrating nutritional and energetic allocation. During fasting, diminished nutritional and energy levels trigger hepatic acceleration of glycogenolysis and gluconeogenesis, while upregulating lipolysis to generate ketone bodies that enhance fatty acid oxidation for systemic energy supply [3]. Prolonged fasting further elevates mitochondrial free radical production, inducing hepatic oxidative stress and lipid peroxidation [4]. Conversely, upon refeeding, elevated nutrient and energy availability augments hepatic glycogenesis while suppressing gluconeogenesis and promoting lipogenesis/secretion [5]. Chronic hyperphagia or sustained high-carbohydrate/high-fat diets induce excessive lipid deposition in the liver, culminating in hepatic steatosis or even steatohepatitis. These conditions subsequently predispose animals to metabolic disorders, ultimately compromising their health and productivity.

The liver is rich in mitochondria, essential organelles within eukaryotic cells, which are critical for maintaining cellular homeostasis. These organelles are responsible for degradation of fatty acids, glucose and amino acids via the tricarboxylic acid cycle and synthesis of ATP through oxidative phosphorylation (OXPHOS). During OXPHOS, electron leakage generates reactive oxygen species (ROS) [6]. Mitochondrial dysfunction can exacerbate ROS production, leading to oxidative stress in the cell [7]. Mitochondria also serve as key carriers of maternal inheritance. Their genetic material, mitochondrial DNA (mtDNA), as a haploid genome, only undergoes minimal DNA recombination. Due to limited protective and repair mechanisms, its mutation rate is typically 10–100 times higher than that of nuclear DNA [8]. Mitochondrial DNA mutations exhibit cumulative effects, where new mutations incorporate all prior mutational information. Consequently, mtDNA from the same maternal lineage clusters into mitochondrial haplogroups. Previous studies have classified domestic chickens (*Gallus gallus domesticus*) and red junglefowl (*Gallus gallus*) into 13 haplogroups (A–I and W–Z) based on mitochondrial D-loop sequences [9]. In the current commercial poultry production of China, the predominant mitochondrial haplogroups in major chicken breeds include A, B, C and E [10]. Among recessive white-feathered chickens and their hybrid offspring, haplogroup E accounts for 48.89% [11]. Since changes in mtDNA sequences may impact the functions of mtDNA encoded genes and mitochondrial-nuclear protein interactions, such changes may further disrupt mitochondrial structure/function and cellular biological processes. Consequently, chickens with different haplogroups exhibit variations in physiological parameters and production traits. Studies indicate that specific mtDNA variants significantly influence traits like breast muscle, fat content and duodenum length in certain chicken breeds [12]. Additionally, broilers carrying mitochondrial haplogroup E demonstrate significantly higher hatch weights and growth rates than those with haplogroups A/B/C [13]. Nevertheless, whether distinct mitochondrial haplogroups exhibit differential responses to nutritional stimuli remains incompletely understood.

This study aims to investigate whether nutritional stimuli affect the structural and functional integrity of mitochondria in hepatocytes and to determine if differential responses to such stimuli exist across distinct haplogroups. These findings will provide mechanistic insights for enhancing animal health and production traits through optimized nutritional/energy modulation.

## 2. Materials and Methods

### 2.1. Experimental Animals

Hens from recessive white-feathered chicken lines with the same genetic background were used as experimental animals. A total of 36 healthy hens at the age of 320 d with similar body weights were randomly selected (18 for A-group and 18 for E-group, their average body weight is 2.690 kg). These experimental animals were provided by the Jiangsu Provincial Institute of Poultry Science. All animal protocols were approved by the Institutional Animal Care and Use Committee of Yangzhou University (SYXK (Su) 2022-0044).

The experimental chickens were randomly divided into fasting and refeeding groups, each containing nine individuals with A or E mitochondrial haplogroups, and the birds were individually housed in cages. The birds in the fasting group were fasted for 12 h, while those in the refeeding group were fasted for 10 h followed by a 2 h feeding period, with free access to feed and water. At the end of the treatment, the chickens were humanely sacrificed by carbon dioxide asphyxiation, and liver tissue samples at the same sites were taken from each chicken, with one part of which was used for the immediate analysis and the left part was snap-frozen in liquid nitrogen and stored at −80 °C for later use.

### 2.2. Isolation and Culture of Primary Hepatocytes

A batch of fertilized eggs with similar sizes produced by the recessive white-feathered hens with A or E mitochondrial haplogroups were selected for incubation, followed by isolating primary hepatocytes from 15-d-old embryos through digestion of liver tissues with 0.2% collagenase IV (Biosharp, Hefei, China), according to the procedures described previously [14]. Primary hepatocytes were plated and cultured in complete culture medium containing Dulbecco’s Modified Eagle Medium (DMEM) (KeyGEN BioTECH, Nanjing, China) supplemented with 10% Fetal Bovine Serum (FBS) (GBICO, Waltham, MA, USA) and 0.02% Epidermal Growth Factor (EGF) (Pepro Tech, Cranbury, NJ, USA). After 24 h of incubation under the condition of 37 °C and 5% CO_2_, the cells were rinsed 3 times with pre-warmed PBS (Solarbio, Beijing, China) before use.

### 2.3. Glucose and Oleic Acid Treatments of Primary Hepatocytes

For glucose treatment, a working solution with a final concentration of 50 mmol/L glucose was made by dissolving glucose (Sigma, Burlington, MA, USA) in the complete medium, which is followed by treating chicken primary hepatocytes with the working solution. The control hepatocytes was only treated with complete medium. For oleic acid treatment, a stock solution of 100 mmol/L oleic acid (Sigma, Burlington, MA, USA) was made by dissolving oleic acid in ethanol, which is followed by mixing the stock solution with the complete medium containing 2% Bovine Serum Albumin (BSA) (Solarbio, Beijing, China) to prepare a working solution with a final concentration of 0.25 mmol/L oleic acid. The hepatocytes were then treated with the working solution. At the same time, the control hepatocytes were cultured in the complete medium supplemented with 2% BSA and the same volume of ethanol. All the hepatocytes were treated for 24 h under the condition of 37 °C and 5% CO_2_. After rinsing the cells with prewarmed PBS, the cells were collected for later use.

### 2.4. Hematoxylin-Eosin (HE) Staining and Oil Red O Staining

The liver tissues were cut into 1 cm × 1 cm × 0.5 cm in size and fixed in 4% Paraformaldehyde Fix Solution (Beyotime, Shanghai, China) at room temperature for 24 h. Subsequently, HE staining and Oil Red O staining were carried out according to the previously described protocols [15].

### 2.5. Transmission Electron Microscopy (TEM)

Liver tissues were cut into small pieces with a sharp blade, placed in microtubes containing 2.5% glutaraldehyde (Yuanye, Shanghai, China), and fixed for 2 h at room temperature in a dark place. Subsequently, the tissue samples were rinsed three times with phosphate buffer containing 1% osmium tetroxide (0.1 mol/L, pH 7.4) for 15 min each time. The tissue samples were then dehydrated sequentially with 30%, 50%, 70%, 80%, 95% and 100% concentrations of alcohol for 20 min per step, which is followed by treating twice with 100% acetone for 15 min each time. The tissue samples were then immersed in a solution of acetone mixed with 812 embedding medium at a ratio of 1:1 and incubated at 37 °C for 2–4 h. Subsequently, the tissue samples were immersed overnight in a solution of acetone mixed with 812 embedding medium in a ratio of 1:2. After that, the tissue samples were placed in pure 812 embedding medium and heated in a 37 °C oven overnight. Next day, the oven temperature was raised to 60 °C to make the embedding medium polymerized with the tissue samples for 48 h, which led to formation of resin block. The resin block was sliced into sections with 60–80 nm thickness, followed by placing the sections on a 150-mesh square copper carrier coated with a carbon film. Subsequently, the carrier was immersed in a 2% uranyl acetate solution prepared in anhydrous ethanol for 8 min. Then, the carrier was rinsed with 70% ethanol three times and with ultrapure water three times, and stained with a 2.6% lead citrate solution without carbon dioxide for 8 min. Finally, the carrier was rinsed three times with ultrapure water, blotted with filter paper, and dried at room temperature overnight. The images of each section were observed and acquired under a transmission electron microscope.

### 2.6. Determination of Triglyceride Content

After 100 mg of liver tissue per sample was accurately weighed, 9 times volume of anhydrous ethanol was added and homogenized, and the supernatant was then collected after centrifugation. Using Triglyceride Assay Kit (Nanjing Jiancheng Bioengineering Institute, Nanjing, China), the triglyceride content in the supernatant was determined according to the manufacturer’s instructions. Similarly, the triglyceride content in cultured cells were also determined after the following procedures: the cells were rinsed 1–2 times with PBS and spinned down at 4 °C and 600× *g* for 10 min, 200 µL of PBS buffer was then added for homogenization and the homogenate was collected for triglyceride determination using the Triglyceride Assay Kit.

### 2.7. Measurement of Mitochondrial Membrane Potential

Mitochondrial membrane potential assay kit with JC-1 (Beyotime, Shanghai, China) was used to determine mitochondrial membrane potential according to the manufacturer’s instructions. In brief, the cells isolated from liver tissues according to the procedures described previously [14] or the cultured liver cells collected were resuspended with the complete medium, followed by adding the JC-1 working solution (made by mixing JC-1 with 1 × JC-1 staining buffer), mixing well and incubating at 37 °C for 20 min. The cells were rinsed with 1 × JC-1 staining buffer 3 times, then the cells was collected by centrifugation, and resuspended with 1 × JC-1 staining buffer. At last, the mitochondrial membrane potential was determined by flow cytometry.

### 2.8. Determination of ROS Level

The ROS level in the cell was determined using the Reactive Oxygen Species Assay Kit (Solarbio, Beijing, China) according to the manufacturer’s instructions. In short, staining solution diluted with Opti-MEM (GBICO, Waltham, MA, USA) was added to the cells isolated from liver tissue or the cultured cells collected, followed by mixing well and incubating at 37 °C for 20 min in a dark place. After incubation, the staining solution was removed, and the cells were rinsed with Opti-MEM 3 times. The cells were then resuspended with an appropriate amount of Opti-MEM. Finally, the ROS level was determined by flow cytometry.

### 2.9. Western Blot Assay

The homogenate of liver tissue or the cultured cells were lysed with radio immunoprecipitation assay (RIPA) lysis buffer (Applygen, Beijing, China) supplemented with protease inhibitor phenylmethanesulfonyl fluoride (PMSF) (Solarbio, Beijing, China) at a ratio of 1:100, which followed by determination of protein concentration using bicinchoninic acid (BCA) protein quantification kit (Vazyme, Nanjing, China) according to the manufacturer’s instructions. As described previously [16], the protein samples were then subjected to electrophoresis on SDS-PAGE (Beyotime, Shanghai, China), followed by transferring the separated proteins to 0.45 μm polyvinylidene difluoride (PVDF) membrane (Merck Millipore, Burlington, MA, USA). Subsequently, the membrane was blocked with NcmBlot blocking buffer (NCM Biotech, Suzhou, China) for 20 min, incubated with primary antibody (1:1000) at 4 °C overnight, and in turn incubated with secondary antibody (1:10,000). At last, the membrane was developed with the Chemiluminescence Kit (Vazyme, Nanjing, China) and imaged using fully automatic chemiluminescence/fluorescence image analysis system (Tanon, Shanghai, China). GoldBand 3-color prestaining protein marker (20351ES, Yeasen, Shanghai, China) was used to indicate protein molecular weight. The primary antibodies used in this study include MFN2 (sc-100560, Santa Cruz Biotechnology, Dallas, TX, USA), β-actin (AC026, ABclonal, Wuhan, China), GAPDH (10494-1-AP, Proteintech, Wuhan, China), SOD2 (24127-1-AP, Proteintech, Wuhan, China), LC3B (3868T, Cell Signaling Technology, Danvers, MA, USA), MT-ND1 (YA2089, MedChemExpress, Monmouth Junction, NJ, USA), CYTB (55090-1-AP, Proteintech, Wuhan, China). The secondary antibodies used in this study include Goat Anti-Rabbit IgG (CW0103S, Cwbiotech, Taizhou, China) and Goat Anti-Mouse IgG (CW01025, Cwbiotech, Taizhou, China).

### 2.10. Statistical Analysis

The obtained data were normalized to the refeeding group or the control group, and Student’s *t*-test was performed to assess the statistical significance of the differences between the fasting and refeeding groups or between the control and treatment groups. Data are presented as the mean ± standard error of the mean (SEM). *p* < 0.05 is set as the criterion for statistical significance.

## 3. Results

### 3.1. Effects of Fasting and Refeeding on Fat Content in the Livers of Chickens with A and E Mitochondrial Haplogroups

Compared to the fasting state, the HE staining showed that the number of vacuoles in the sections of chicken livers in both A and E groups was higher in the refeeding state (*p* < 0.05); oil Red O staining showed that the fat content in chicken livers in both A and E groups was higher in the refeeding state (*p* < 0.05); triglyceride content measurements indicated that triglyceride content in chicken livers in both A and E groups was higher in the fed state (*p* < 0.05) (Figure 1A–E).

### 3.2. Effects of Fasting and Refeeding on the Mitochondrial Number, Structure and Morphology in the Livers of Chickens with A and E Mitochondrial Haplogroups

Compared to the fasting state, TEM analysis indicated that for the A group, there were no significant differences in the average mitochondrial circumference, average mitochondria-associated endoplasmic reticulum membrane (MAM) length, average mitochondrial number and average mitochondrial area in the refeeding state; however, the number of aggregated ribosomes was higher in the refeeding state (*p* < 0.05) (Figure 2A–H). For the E group, there were no significant differences in the average mitochondrial circumference, average MAM length and average mitochondrial area in the refeeding state; however, the number of mitochondria was lower (*p* < 0.05) and the number of aggregated ribosomes was higher (*p* < 0.05) in the refeeding state (Figure 3A–H).

### 3.3. Effects of Fasting and Refeeding on Mitochondrial Membrane Potential and ROS Level in the Livers of Chickens with A and E Mitochondrial Haplogroups

Compared with the fasting state, the mitochondrial membrane potential level in the A group was higher in the refeeding state (*p* < 0.05), but there was no significant difference in ROS level between the fasting and refeeding states (Figure 4A,B). Differently, compared to the fasting state, both the mitochondrial membrane potential level and ROS level in the E group were higher in the refeeding state (*p* < 0.05) (Figure 5A,B).

### 3.4. Effects of Fasting and Refeeding on the Abundance of Mitochondria-Associated Proteins in the Livers of Chickens with A and E Mitochondrial Haplogroups

The immunoblotting analysis indicated that, compared to the fasting state, MFN2 and SOD2 protein contents in the livers of chickens with A mitochondrial haplogroup were lower in the refeeding state (*p* < 0.05), but there were no significant differences in the protein contents of CYTB, MT-ND1 and LC3B between the fasting and refeeding states. (Figure 6A–F). For the E mitochondrial haplogroup, there were no significant differences in the contents of any of the abovementioned proteins between the fasting and refeeding states (Figure 7A–F).

### 3.5. Effects of Glucose and Oleic Acid Treatments on Mitochondrial Membrane Potential and ROS Level in Primary Hepatocytes with A and E Mitochondrial Haplogroups

Compared to the control group, the mitochondrial membrane potential was higher in the glucose-treated primary chicken hepatocytes with the A mitochondrial haplogroup (*p* < 0.01), but there was no significant difference in ROS level between the control and treated groups (Figure 8A,B). Differently, there was no significant differences in the mitochondrial membrane potential and the ROS level between the control and treated groups of chicken hepatocytes with the E mitochondrial haplogroup (Figure 9A,B). For oleic acid treatment, compared to the control group, the mitochondrial membrane potential was lower in the treated primary chicken hepatocytes with the A mitochondrial haplogroup (*p* < 0.05), and there was no significant difference in the ROS level between the control and treated groups (Figure 10A,B). Differently, there was no significant difference in mitochondrial membrane potential between the control and treated groups of primary chicken hepatocytes with the E mitochondrial haplogroup, but the ROS level was lower in the treated group than the control group (*p* < 0.01) (Figure 11A,B).

### 3.6. Effects of Glucose and Oleic Acid Treatments on the Abundance of Mitochondria-Related Proteins in Primary Hepatocytes with A and E Mitochondrial Haplogroups

The immunoblotting analysis indicated that glucose treatment had no significant effect on the protein contents of MFN2, CYTB, MT-ND1, SOD2 and LC3B in primary chicken hepatocytes with either A or E mitochondrial haplogroups (Figure 12A–L). For oleic acid treatment, the protein content of CYTB in primary chicken hepatocytes with the A mitochondrial haplogroup was significantly higher in the treated group than that of the control group (*p* < 0.05), but there were no significant differences in the protein contents of MFN2, MT-ND1, SOD2 and LC3B between the control and treated groups (Figure 13A–F). Differently, there were no significant differences in the contents of any of the abovementioned proteins between the control group and treated groups of primary chicken hepatocytes with the E mitochondrial haplogroup (Figure 13G–L).

### 3.7. Effect of Glucose and Oleic Acid Treatments on the Fat Content of Primary Hepatocytes with A and E Mitochondrial Haplogroups

Triglyceride content analysis indicated that compared to the control group, the triglyceride contents in the chicken primary hepatocytes with both A and E mitochondrial haplogroups were not significantly different between the control and treated groups in response to glucose treatment (Figure 14A). Differently, for oleic acid treatment, the triglyceride contents of the chicken primary hepatocytes with both A and E mitochondrial haplogroups were higher in the treated group than the control group (*p* < 0.01) (Figure 14B).

## 4. Discussion

Appropriate nutritional supply promotes the growth, health and production performance of livestock and poultry. This study utilizes in vivo fasting/refeeding models and in vitro cell models treated with different nutritional factors to elucidate the impacts of nutritional/energy disturbances on mitochondrial structure and function, oxidative stress and fat deposition in liver cells. The findings from this study may lay a foundation for revealing the mechanisms by which nutritional/energy levels regulate the growth, health and production performance of livestock and poultry. The results indicated that feeding significantly increased hepatic lipid deposition in chickens compared to fasting. Hepatic triglyceride deposition may originate from either glucose conversion via the tricarboxylic acid (TCA) cycle in the mitochondrial matrix or direct deposition of exogenous fat [17,18]. Consistently, treatments of chicken primary hepatocytes with 50 mmol/L glucose or 0.25 mmol/L oleic acid also increased fat deposition. Taken together, in vivo and in vitro studies indicated that nutrition/energy surplus promoted lipid accumulation in hepatocytes; however, this phenomenon was not affected by mitochondrial haplogroups.

Previous studies have provided some mechanistic explanations for feeding to promote hepatic fat deposition. During fasting, the liver mobilizes fat to supply energy, concurrently activating AMPK to suppress the expression of genes encoding glycogen synthase and FASN [19]. Meanwhile, mTORC2 signaling pathway is activated, initiating mitochondrial fission to maintain respiratory chain stability and enhance fatty acid β-oxidation by increasing mitochondrial quantity [20]. Upon refeeding, AMPK activity is inhibited, reducing mitochondrial β-oxidation and reactivating fatty acid synthesis pathways. Consequently, feeding promotes hepatic lipid deposition.

In the feeding state, when nutrients or energy influx into the cell, the cell needs to synthesize a large number of enzymes, signaling molecules and other proteins to participate in the metabolism of nutrients [3], so that a larger number of protein synthesis factories-ribosomes are seen to be aggregated together under electron microscopy. In addition, the level of oxidative phosphorylation involved in the electron transport chain on the inner mitochondrial membrane and the mitochondrial membrane potential were elevated during the influx of nutrients and energy into the cell without influence of mitochondrial haplogroup. The increase in mitochondrial membrane potential in the refeeding state can be partially attributed to the large influx of glucose, but not fat, into the cell. This is consistent with the findings that glucose treatment resulted in increase of mitochondrial membrane potential, whereas oleic acid treatment resulted in decrease of mitochondrial membrane potential in chicken liver cells with the A mitochondrial haplogroup. Since glucose needs to go through the TAC in the mitochondrial matrix to be converted to fat, and the conversion process requires a large amount of ATP, the cells need to increase the level of oxidative phosphorylation and mitochondrial membrane potential to synthesize a large amount of ATP. Unlike glucose, oleic acid does not need to pass through the TAC to be converted to fat, and does not need to raise or even lower oxidative phosphorylation level and mitochondrial membrane potential. In addition, it is worth mentioning that fatty acids synthesized directly from glucose are saturated fatty acids, whereas oleic acid is an unsaturated fatty acid, and previous studies have shown that saturated fatty acids and unsaturated fatty acids have quite different biological effects [21,22]. Unsaturated fatty acids, such as oleic and linoleic acids, inhibit cytotoxic effects such as saturated fatty acid-induced oxidative stress and unsaturated fatty acids can alter cell membrane and mitochondrial membrane, thus affecting the structure and function of mitochondria [23].

The elevated level of oxidative phosphorylation is usually accompanied by increased electron leakage from the electron transport chain on the inner mitochondrial membrane, which in turn increases the cellular ROS level [24]. Consistent with this, the ROS level in chicken hepatocytes with the E mitochondrial haplogroup was significantly higher in the refeeding state than that in the fasting state, and the ROS level in oleic acid-treated primary chicken hepatocytes with the E mitochondrial haplogroup was significantly lower than that in control hepatocytes. Furthermore, ROS level in A-group hepatocytes was not significantly higher in the refeeding state than in the fasting state, which reflects that A-group hepatocytes have stronger adaptive capacity in response to nutrient/energy stimuli, such as stronger metabolic regulation or stronger antioxidant capacity. The aforementioned difference in ROS level between the A and E groups in response to refeeding also suggests that mitochondrial haplogroups have an effect on the cells’ ability to resist oxidative stress.

The elevated level of cellular ROS usually results in oxidation of mitochondrial lipids and proteins, causing damage to mitochondrial structure and function [7]. Cells have pathways to remove damaged mitochondria, such as mitochondrial autophagy, so elevated ROS level may lead to a reduction in the number of mitochondria. Indeed, the number of mitochondria in the E group hepatocytes was significantly reduced in the refeeding state relative to the fasting state, and this reduction may be related to the increased mitochondrial damage induced by elevated ROS level. Consistent with this, neither ROS level nor the number of mitochondria in the A group hepatocytes changed significantly during the refeeding state relative to the fasting state.

As for the significantly lower protein contents of MFN2 and SOD2 in hepatocytes with the A mitochondrial haplogroup in the refeeding state than in the fasting state, this may be related to the activation of the AMPK signaling pathway during fasting. Previous studies have reported that when the cellular energy level is low such as in the fasting state, the activation of AMPK can promote the fusion of mitochondria and the activation of KEAP1/NRF2 signaling pathway [25,26], while MFN2 is a key protein mediating the fusion of mitochondria [27], and KEAP1/NRF2, as a pivot to regulate the antioxidative stress, can promote the expression of antioxidant enzyme SOD2, so that the fasting state of A-group hepatocytes would have significantly higher protein contents of MFN2 and SOD2 than the refeeding state. However, this was not the case for E-group and the protein contents of MFN2 and SOD2 were not significantly different between the refeeding and fasting states. This suggests that mitochondrial fusion and antioxidant capacities differ between mitochondrial haplogroups and that the AMPK pathway may be insensitive to fasting in the E-group hepatocytes, although this needs to be verified in further studies.

In the present study, we also found that oleic acid treatment induced the expression of CYTB protein, a core component of electron transport in the inner mitochondrial membrane [28], which may help to reduce the leakage of electrons from the electron transport chain during oxidative phosphorylation and reduce the level of ROS [29]. As to how oleic acid induces CYTB expression, the relevant mechanism is still unclear, which remains to be elucidated in future studies. However, the inducing effect of oleic acid on CYTB expression was not evident in the E-group hepatocytes, suggesting that the inducing effect of oleic acid on CYTB expression is also affected by the mitochondrial haplogroups. In addition, some of the above indicators, such as mitochondrial membrane potential, ROS level and protein contents of MFN2, SOD2 and CYTB, differed between glucose and oleic acid treatments.

The findings from this study have some implications for animal production. Firstly, we found that feeding promoted fat deposition in the liver of chicken; thus, overfeeding may lead to obesity and obesity-associated diseases such as non-alcoholic fatty liver disease. In practice, the feed formula and intake should match the actual nutritional needs of animals to avoid excessive nutrient or energy intake. Secondly, we found that during fasting, the oxidative phosphorylation level in the cell was low, which could lead to insufficient energy required for their life activities, affecting their growth and production performance. Therefore, when animals are transported over long distances and suffer from nutritional deficiencies, it is better to provide them with energy in advance and replenish food promptly upon arrival at their destination. Thirdly, we found that the number of clustered ribosomes in cells increased during feeding, which reflects that feeding promotes protein synthesis in cells and animal growth. Therefore, in practice, the protein level and composition of the diet should be reasonable and balanced, which will be beneficial for the growth and development of animals. Fourthly, we found that compared to the A haplogroup, the E haplogroup was more prone to oxidative stress during feeding. This suggests that in practice, different amounts of antioxidants should be supplemented according to the mitochondrial haplogroups among different breeds of animals. For example, the breeds with E haplogroup should add more antioxidants than the breeds with A haplogroup. Lastly, we found that adding glucose was beneficial to mitochondrial biogenesis of A haplogroup, while adding oleic acid did not induce oxidative stress in E haplogroup. This suggests that in practice, breeds with A haplogroup are more suitable for carbohydrate-based dietary formula, while adding oleic acid appropriately is more beneficial to the breeds with E haplogroup.

In this study, we selected 50 mmol/L glucose and 0.25 mmol/L oleic acid to treat primary hepatocytes, slightly above physiological concentrations. This was primarily based on the following considerations: Firstly, in vivo effects are typically chronic, thus requiring higher concentrations to achieve the same effects within a short period of treatment in vitro. Secondly, the concentrations of specific substances in the blood of healthy animals are generally considered their physiological concentrations, and the intracellular concentrations of the substances may far exceed the physiological levels due to cellular active transportation, thus treating cells with higher concentrations than physiological levels may better mimic the conditions in vivo. Thirdly, using higher concentrations during cell treatment may be more conducive to revealing relevant molecular mechanisms. Lastly, the concentrations in the formal cell experiment also took into account the result of a preliminary study.

It is noteworthy that this study has the following limitation: the chickens used in the in vivo experiment were 320 d old, but the cells used in the in vitro experiment were not isolated from 320-d-old chickens, but from chicken embryos. This inconsistency may lead to differences in the results. It is necessary to validate the results by treating cells isolated from chickens of the same age after solving technical problems.

## 5. Conclusions

Nutritional and energy disturbances can affect mitochondrial structure and function, oxidative stress, and fat deposition in cells, and some of the effects also depend on mitochondrial haplogroups and the physical and chemical properties of nutritional factors. Based on these findings, it is necessary to supplement energy appropriately according to the nutritional status of animals, avoid excessive feeding, scientifically prepare feed (e.g., supplementing balanced protein or appropriate amount of antioxidants to feed), and consider the differences in mitochondrial haplogroups among different breeds of animals.

## Figures and Tables

**Figure 1 animals-15-03151-f001:**
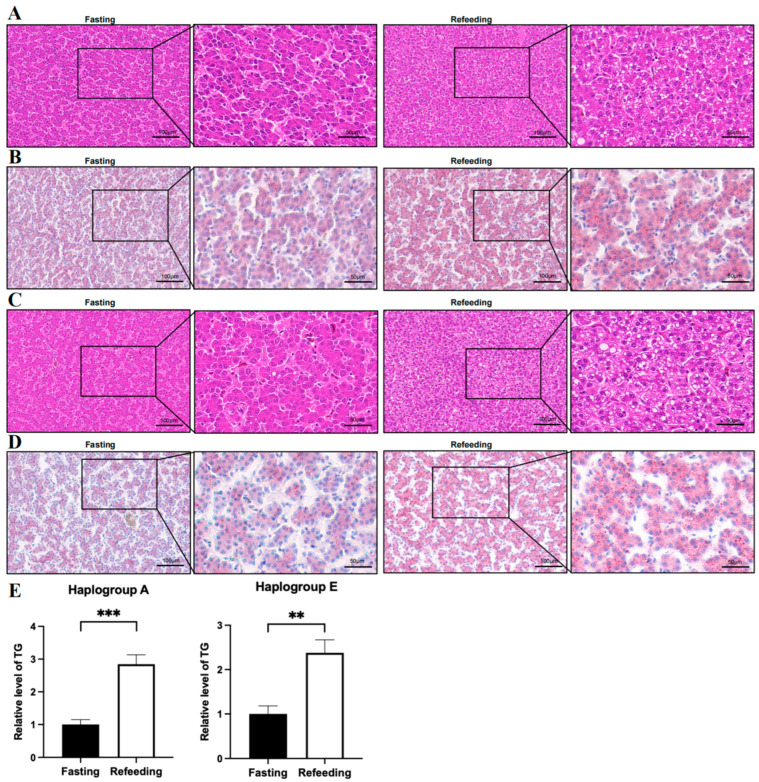
Effects of fasting and refeeding on the intracellular fat content of chicken livers with mitochondrial haplogroups A and E. (**A**) Representative images of HE staining analysis on the livers of chickens with mitochondrial haplogroup A in the fasting and refeeding states. (**B**) Representative images of Oil Red O staining analysis on the livers of chickens with mitochondrial haplogroup A in the fasting and refeeding states. (**C**) Representative images of HE staining analysis on the livers of chickens with mitochondrial haplogroup E in the fasting and refeeding states. (**D**) Representative images of Oil Red O staining analysis on the livers of chickens with mitochondrial haplogroup E in the fasting and refeeding states. Scale bars are 100 μm or 50 μm. (**E**) Histograms showing the relative content triglycerides in the livers of chickens with mitochondrial haplogroups A and E in the fasting and refeeding states, *n* = 9. Data are presented as the mean ± SEM. **, *** denote *p* < 0.01 and 0.001, respectively.

**Figure 2 animals-15-03151-f002:**
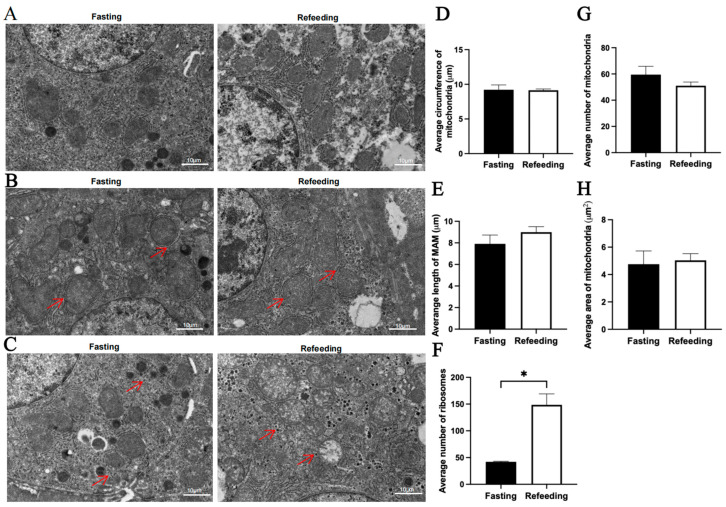
Effects of fasting and refeeding on the number, structure, and morphology of mitochondria in the livers of chickens with mitochondrial haplogroup A. (**A**) Representative TEM images of mitochondria in the livers of chickens with mitochondrial haplogroup A in the fasting and refeeding states. (**B**) Representative TEM images of MAM in the livers of chickens with mitochondrial haplogroup A. Red arrows point to MAM. (**C**) Representative TEM images of aggregated ribosomes in the liver cells of chickens with mitochondrial haplogroup A. Red arrows point to the aggregated ribosomes. (**D**) Histograms showing the average circumference of mitochondria in liver cells of chickens with mitochondrial haplogroup A. Six TEM images were selected per chicken, and each image was used to count the number of circular mitochondria, quantify the total circumference of circular mitochondria, and calculate the average mitochondrial circumference. (**E**) Histograms showing the average MAM length in hepatocytes of chickens with mitochondrial haplogroup A. Six TEM images were selected per chicken, and each image was used to quantify the total MAM length, to count the number of mitochondria surrounded by MAM, and to calculate the average MAM length. (**F**) Histograms showing the average number of aggregated ribosomes in the liver cells of chickens with mitochondrial haplogroup A. Six TEM images were selected per chicken and each image was used to count the number of aggregated ribosomes. (**G**,**H**) The histograms showing the average number of mitochondria and the average area of mitochondria in hepatocytes of chickens with mitochondrial haplogroup A. The average number of mitochondria per chicken was calculated by dividing the total number of mitochondria in the 12 images by 2, the average mitochondrial area for each group was calculated by dividing the total area of mitochondria in the 12 images by the total number of mitochondria. Scale bars for (**A**–**C**) are 10 μm. *n* = 2. Data are presented as the mean ± SEM. * denotes *p* < 0.05.

**Figure 3 animals-15-03151-f003:**
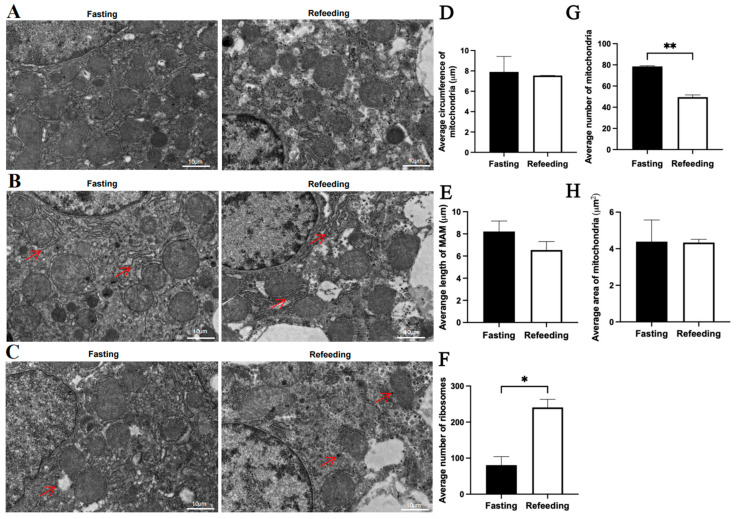
Effects of fasting and feeding on the number, structure, and morphology of mitochondria in the livers of chickens with mitochondrial haplogroup E. (**A**) Representative TEM images of mitochondria in the livers of chickens with mitochondrial haplogroup E in the fasting and refeeding states. (**B**) Representative TEM images of MAM in the livers of chickens with mitochondrial haplogroup E. Red arrows point to MAM. (**C**) Representative TEM images of aggregated ribosomes in the liver cells of chickens with mitochondrial haplogroup E in the fasting and refeeding states. Red arrows point to the aggregated ribosomes. (**D**) Histograms showing the average circumference of mitochondria in liver cells of chickens with mitochondrial haplogroup E. (**E**) Histograms showing the average MAM length in hepatocytes of chickens with mitochondrial haplogroup E. (**F**) Histograms showing the average number of aggregated ribosomes in the liver cells of chickens with mitochondrial haplogroup E. (**G**,**H**) The histograms showing the average number of mitochondria and the average area of mitochondria in hepatocytes of chickens with mitochondrial haplogroup E. Scale bars for (**A**–**C**) are 10 μm. *n* = 2. Data are presented as the mean ± SEM. *, ** denote *p* < 0.05 and 0.01, respectively.

**Figure 4 animals-15-03151-f004:**
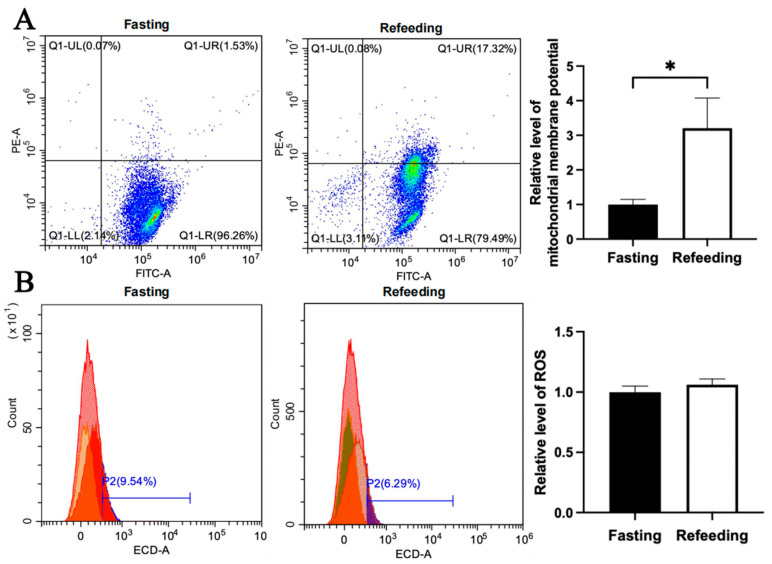
Effects of fasting and feeding on mitochondrial membrane potential and ROS level in the livers of chickens with mitochondrial haplogroup A. (**A**) Representative images showing flow cytometry analysis of mitochondrial membrane potential and histograms showing relative level of mitochondrial membrane potentials in liver cells of chickens with mitochondrial haplogroup A in the fasting and refeeding states. The X-axis and Y-axis indicate the number of cells emitting green fluorescence and red fluorescence, respectively. Mitochondrial membrane potential was determined by the ratio of red to green fluorescence, n = 9. (**B**) Representative images showing flow cytometry analysis of ROS level and histograms showing the relative level of ROS in liver cells of chickens with mitochondrial haplogroup A in the fasting and refeeding states. The X-axis and Y-axis indicate the intensity of red fluorescence and the number of liver cells, respectively. *n* = 9. Data are presented as the mean ± SEM. * denotes *p* < 0.05.

**Figure 5 animals-15-03151-f005:**
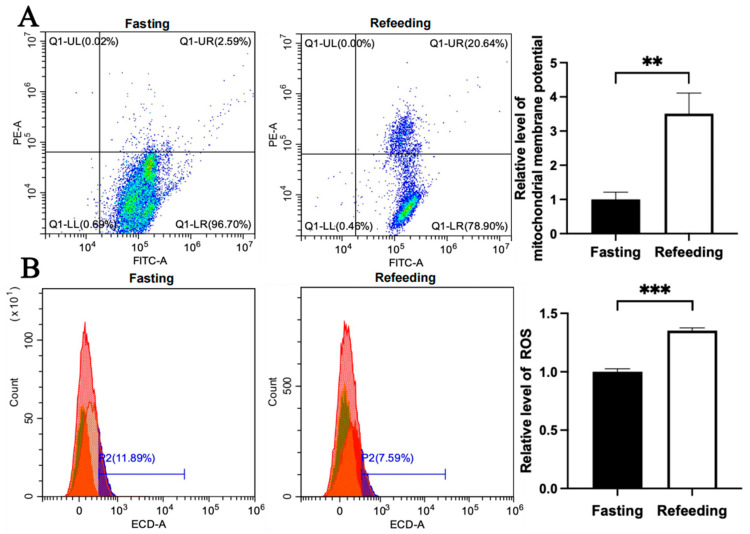
Effects of fasting and feeding on mitochondrial membrane potential and ROS level in the livers of chickens with mitochondrial haplogroup E. (**A**) Representative images showing flow cytometry analysis of mitochondrial membrane potential and histograms showing the relative level of mitochondrial membrane potentials in liver cells of chickens with mitochondrial haplogroup E in the fasting and refeeding states. The X-axis and Y-axis indicate the number of cells emitting green fluorescence and red fluorescence, respectively. Mitochondrial membrane potential was determined by the ratio of red to green fluorescence, n = 9. (**B**) Representative images showing flow cytometry analysis of ROS level and histograms showing the relative level of ROS in liver cells of chickens with mitochondrial haplogroup E in the fasting and refeeding states. The X-axis and Y-axis indicate the intensity of red fluorescence and the number of liver cells, respectively, *n* = 9. Data are presented as the mean ± SEM. **, *** denote *p* < 0.01 and 0.001, respectively.

**Figure 6 animals-15-03151-f006:**
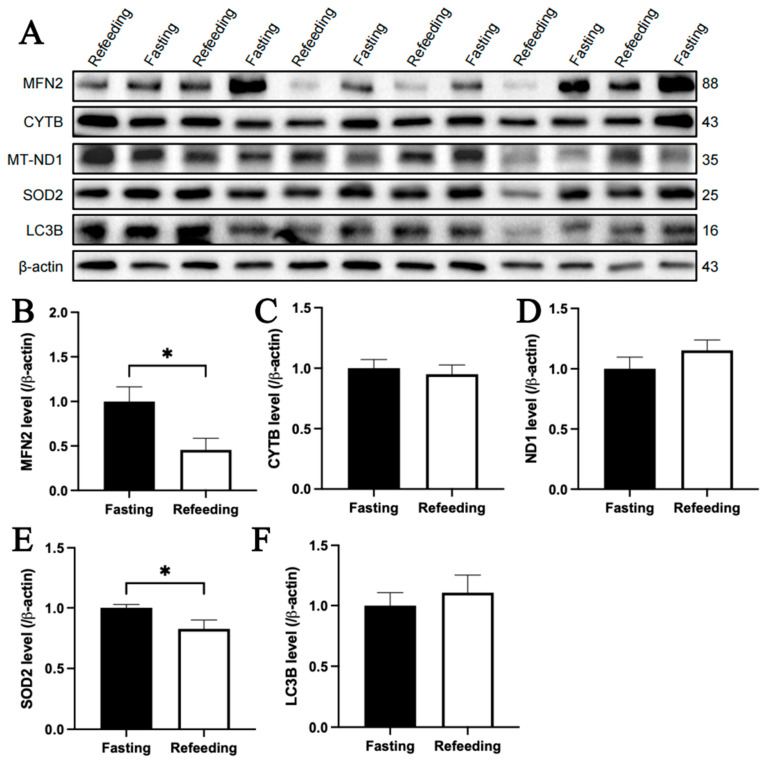
Effects of fasting and feeding on the content of mitochondria-related proteins in the livers of chickens with mitochondrial haplogroup A. (**A**) Representative immunoblot images of mitochondria-related proteins in the livers of chickens with mitochondrial haplogroup A in the fasting and refeeding states. (**B**–**F**) The histograms showing quantification of the immunoblots of mitochondria-related proteins in the livers of chickens with mitochondrial haplogroup A in the fasting and refeeding states. β-actin was used as an internal reference gene. All the samples were obtained from the same experiment or parallel experiments, and the blots were processed in parallel. *n* = 6. Data are presented as the mean ± SEM. * denotes *p* < 0.05.

**Figure 7 animals-15-03151-f007:**
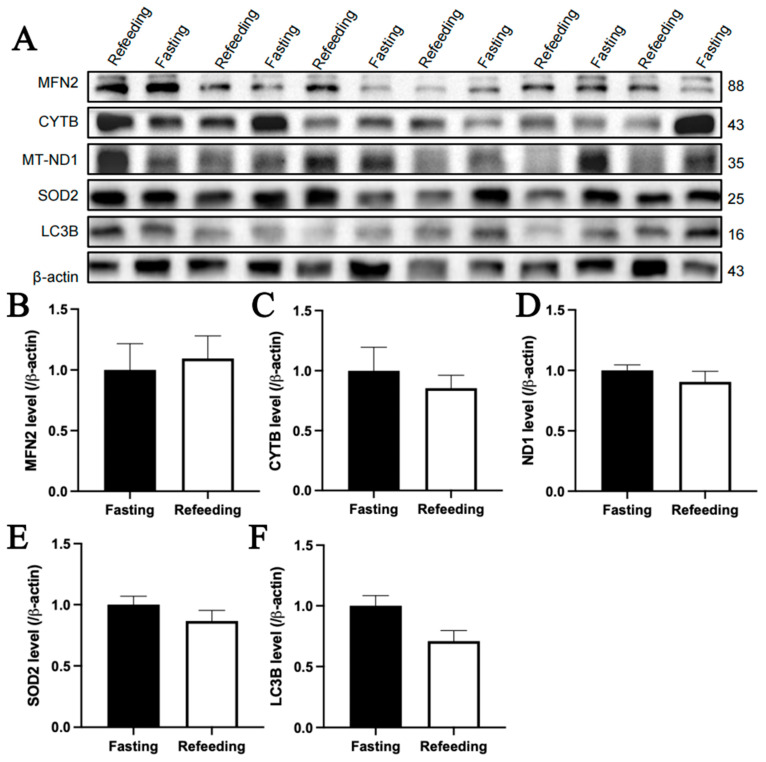
Effects of fasting and feeding on the content of mitochondria-related proteins in the livers of chickens with mitochondrial haplogroup E. (**A**) Representative immunoblot images of mitochondria-related proteins in the livers of chickens with mitochondrial haplogroup E in the fasting and refeeding states. (**B**–**F**) The histograms showing quantification of the immunoblots of mitochondria-related proteins in the livers of chickens with mitochondrial haplogroup E in the fasting and refeeding states. β-actin was used as an internal reference gene. All the samples were obtained from the same experiment or parallel experiments, and the blots were processed in parallel. *n* = 6. Data are presented as the mean ± SEM.

**Figure 8 animals-15-03151-f008:**
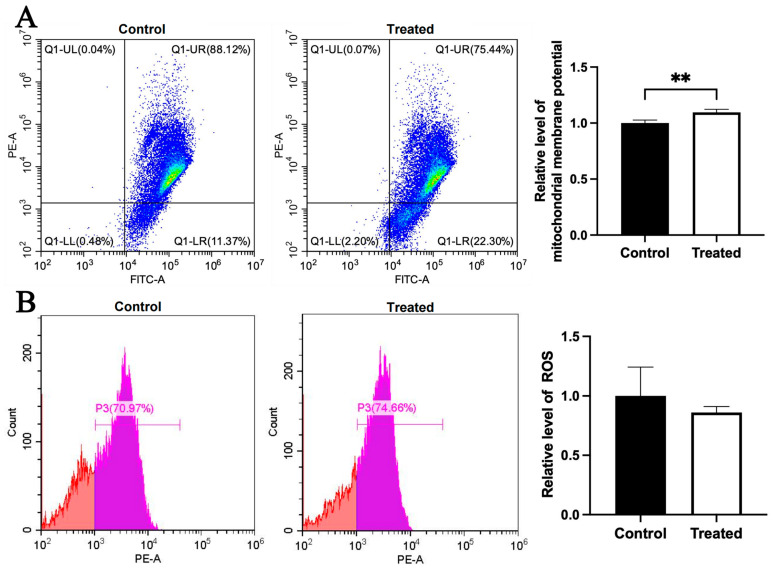
Effects of glucose treatment on mitochondrial membrane potential and ROS level in the primary hepatocytes of chickens with mitochondrial haplogroup A. (**A**) Representative images showing flow cytometry analysis of mitochondrial membrane potential and histograms showing relative level of membrane potential in the control and glucose-treated primary hepatocytes of chickens with mitochondrial haplogroup A. The X-axis and Y-axis indicate the number of cells emitting green and red fluorescence, respectively. Mitochondrial membrane potential was determined by the ratio of red to green fluorescence. n = 4. (**B**) Representative images showing flow cytometry analysis of ROS level and histograms showing the relative level of ROS in the control and glucose-treated primary hepatocytes of chickens with mitochondrial haplogroup A. The X-axis and Y-axis indicate the intensity of red fluorescence and the number of cells, respectively. *n* = 4. Data are presented as the mean ± SEM. ** denotes *p* < 0.01.

**Figure 9 animals-15-03151-f009:**
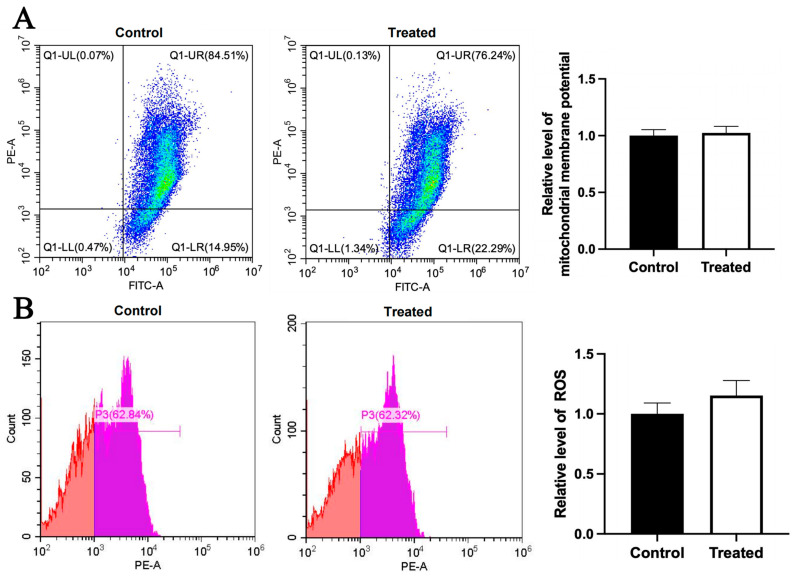
Effects of glucose treatment on mitochondrial membrane potential and ROS level in the primary hepatocytes of chickens with mitochondrial haplogroup E. (**A**) Representative images showing flow cytometry analysis of mitochondrial membrane potential and histograms showing relative level of membrane potential in the control and glucose-treated primary hepatocytes of chickens with mitochondrial haplogroup E. The X-axis and Y-axis indicate the number of cells emitting green and red fluorescence, respectively. Mitochondrial membrane potential was determined by the ratio of red to green fluorescence. n = 4. (**B**) Representative images showing flow cytometry analysis of ROS level and histograms showing the relative level of ROS in the control and glucose-treated primary hepatocytes of chickens with mitochondrial haplogroup E. The X-axis and Y-axis indicate the intensity of red fluorescence and the number of cells, respectively. *n* = 4. Data are presented as the mean ± SEM.

**Figure 10 animals-15-03151-f010:**
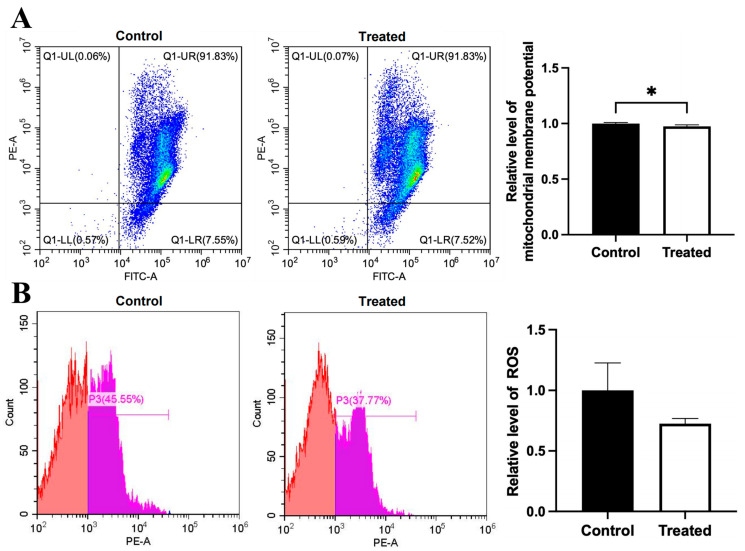
Effects of oleic acid treatment on mitochondrial membrane potential and ROS level in the primary hepatocytes of chickens with mitochondrial haplogroup A. (**A**) Representative images showing flow cytometry analysis of mitochondrial membrane potential and histograms showing relative level of membrane potential in the control and oleic acid-treated primary hepatocytes of chickens with mitochondrial haplogroup A. The X-axis and Y-axis indicate the number of cells emitting green and red fluorescence, respectively. Mitochondrial membrane potential was determined by the ratio of red to green fluorescence. n = 4. (**B**) Representative images showing flow cytometry analysis of ROS level and histograms showing the relative level of ROS in the control and oleic acid-treated primary hepatocytes of chickens with mitochondrial haplogroup A. The X-axis and Y-axis indicate the intensity of red fluorescence and the number of cells, respectively. *n* = 4. Data are presented as the mean ± SEM. * denotes *p* < 0.05.

**Figure 11 animals-15-03151-f011:**
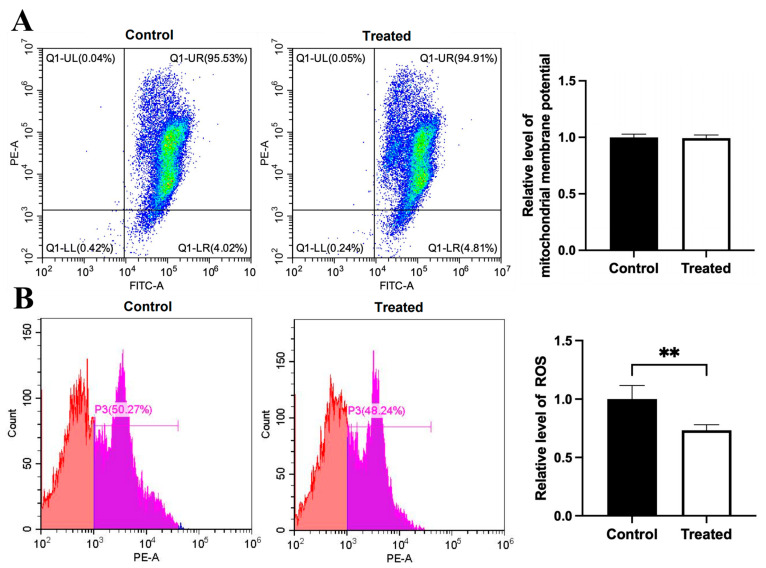
Effects of oleic acid treatment on mitochondrial membrane potential and ROS level in the primary hepatocytes of chickens with mitochondrial haplogroup E. (**A**) Representative images showing flow cytometry analysis of mitochondrial membrane potential and histograms showing relative level of membrane potential in the control and oleic acid-treated primary hepatocytes of chickens with mitochondrial haplogroup E. The X-axis and Y-axis indicate the number of cells emitting green and red fluorescence, respectively. Mitochondrial membrane potential was determined by the ratio of red to green fluorescence. n = 4. (**B**) Representative images showing flow cytometry analysis of ROS level and histograms showing the relative level of ROS in the control and oleic acid-treated primary hepatocytes of chickens with mitochondrial haplogroup E. The X-axis and Y-axis indicate the intensity of red fluorescence and the number of cells, respectively. *n* = 4. Data are presented as the mean ± SEM. ** denotes *p* < 0.01.

**Figure 12 animals-15-03151-f012:**
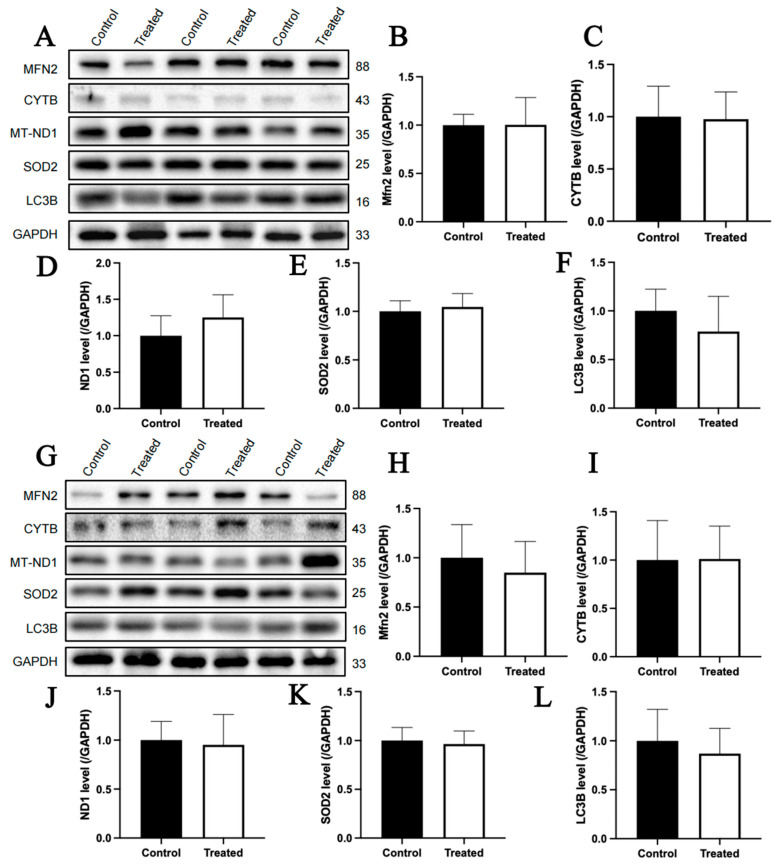
Effects of glucose treatment on mitochondria-related protein content in the primary hepatocytes of chickens with mitochondrial haplogroups A and E. (**A**) Representative images showing the immunoblots of mitochondria-related protein content in the control and glucose-treated primary hepatocytes of chickens with mitochondrial haplogroup A. (**B**–**F**) The histograms showing quantification of the immunoblots of mitochondria-related proteins in the control and glucose-treated primary hepatocytes of chickens with mitochondrial haplogroup A. (**G**) Representative images showing the immunoblots of mitochondria-related protein content in the control and glucose-treated primary hepatocytes of chickens with mitochondrial haplogroup E. (**H**–**L**) The histograms showing quantification of the immunoblots of mitochondria-related proteins in the control and glucose-treated primary hepatocytes of chickens with mitochondrial haplogroup E. GAPDH was used as an internal reference gene. All the samples were obtained from the same experiment or parallel experiments, and the blots were processed in parallel. *n* = 3. Data are presented as the mean ± SEM.

**Figure 13 animals-15-03151-f013:**
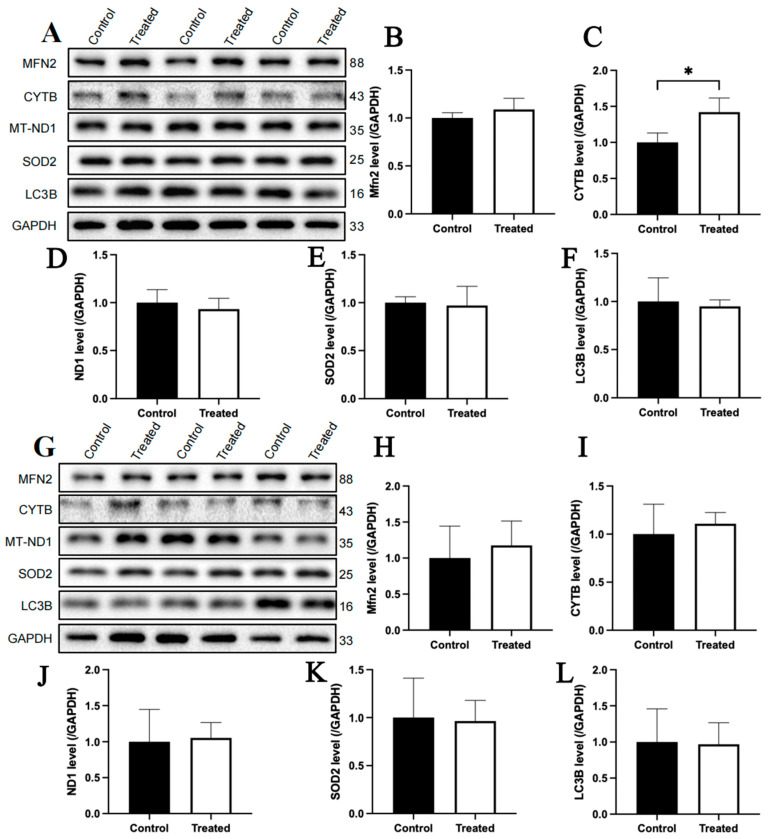
Effects of oleic acid treatment on the content of mitochondria-related proteins in the primary hepatocytes of chickens with mitochondrial haplogroups A and E. (**A**) Representative images showing the immunoblots of mitochondria-related protein content in the control and oleic acid-treated primary hepatocytes of chickens with mitochondrial haplogroup A. (**B**–**F**) The histograms showing quantification of the immunoblots of mitochondria-related proteins in the control and oleic acid-treated primary hepatocytes of chickens with mitochondrial haplogroup A. (**G**) Representative images showing the immunoblots of mitochondria-related protein content in the control and oleic acid-treated primary hepatocytes of chickens with mitochondrial haplogroup E. (**H**–**L**) The histograms showing quantification of the immunoblots of mitochondria-related proteins in the control and oleic acid-treated primary hepatocytes of chickens with mitochondrial haplogroup E. GAPDH was used as an internal reference gene. All the samples were obtained from the same experiment or parallel experiments, and the blots were processed in parallel. *n* = 3. Data are presented as the mean ± SEM. * denotes *p* < 0.05.

**Figure 14 animals-15-03151-f014:**
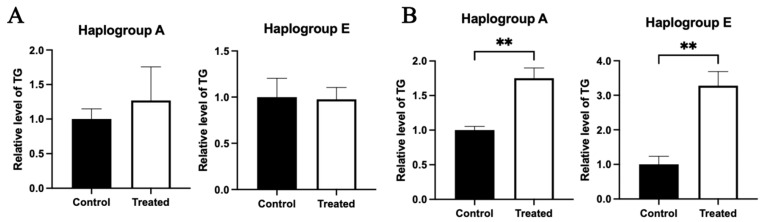
Effects of glucose or oleic acid treatments on the intracellular fat content in the primary hepatocytes of chickens with mitochondrial haplogroups A and E. (**A**) The histograms showing triglyceride content in the control and glucose-treated primary hepatocytes of chickens with mitochondrial haplogroup A and E, respectively. (**B**) The histograms showing triglyceride content in the control and oleic acid-treated primary hepatocytes of chickens with mitochondrial haplogroup A and E, respectively. *n* = 3. Data are presented as the mean ± SEM. ** denotes *p* < 0.01.

## Data Availability

We confirm that all data supporting the findings in this study are available within the article.

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
