# Peer review of "Effects of Nutritional Disturbances on the Structure and Function of Mitochondria, Oxidative Stress Level, and Fat Deposition in Chicken Liver Cells"

_animals, 2025, doi:10.3390/ani15213151_

Round 1
Reviewer 1 Report
Comments and Suggestions for Authors
Brief Summary
The study investigated the effect of fasting/refeeding of broiler chicken from 2 different mtDNA haplogroups on various hepatic memetabolism parameters. This concept is interesting and original.
The experimental design appears to be correct, and the methodology seems appropriate.The techniques used, seem as far as I can judge to be well suited.
The study is generally well written in a correct English except the abstract.
On the other hand, numerous conceptual and editorial defects raise questions for me.
General Concept
The abstract is confusing and absolutely needs to be rewritten.
Firstly, it would have been interesting to combine the hepatic approach with results on zootechnical performance and body condition and fat deposition of the carcasses. If you have them, please include them in the paper.
Secondly, your study is in a 2 x 2 factorial design (haplogroups x fasting/refeeding), yet your statistical analysis is very basic (Student's t-test). By doing this, you lose statistical power, you cannot measure potential interactions between treatments, and moreover, the presentation of the results is extremely cumbersome and boring. I recognize that this may be a problem for images, but I find that this statistical analysis is really not up to the quality of your results. Could you review it ?
Also you mentionned in the discussion The findings from this study may lay a foundation for revealing the mechanisms by which nutritional/energy levels regulat the growth, health and production performance of livestock and poultry. But in the conclusion, you do not clearly mention the practical implications of your results. Should these fasting-refeeding systems be used in farming? How? Please provide some more applied elements in your general conclusions.
This article absolutely needs to be reviewed and reorganized.
Comments on the Quality of English LanguageThe abstract is confusing and absolutely needs to be rewritten.
Line 137 I would suggest « sacrified » instead of « executed » .
Reviewer 2 Report
Comments and Suggestions for Authors
This study focuses on the interaction between nutritional fluctuations and chicken mitochondrial haplotypes (Types A and E). By employing an in vivo fasting/refeeding model and in vitro experiments with glucose and oleic acid treatments, it systematically investigates the mechanisms underlying changes in hepatocyte mitochondrial function, oxidative stress, and lipid deposition. The research topic aligns with the current research hotspots in poultry nutritional metabolism and molecular breeding, and possesses clear theoretical value and application potential. However, the manuscript needs improvement in aspects such as the depth of mechanism explanation. Specific comments are as follows:
1. Should animal feeding experiments be adopted to study the effect of nutritional regulation? This would be more in line with the scope of this journal.
2. In Section 1.1 of the Introduction, the data "the total chicken production reached 26.37 million tons in 2024" is not marked with a data source; it is recommended to supplement the data source.
3. L50: Please change the ‘CO2’ to ‘CO2’.
4. In Section 2.1, although the in vivo experiment clearly states that there are 18 chickens in each of the haplotype A and E groups (9 in the fasting group and 9 in the refeeding group), "n=2" is indicated in the annotation of transmission electron microscopy (TEM) analysis in Section 3.2. How to determine the number of samples for analysis? It is recommended to supplement the basis for selecting TEM samples.
5. In the in vitro experiment in Section 2.3, the concentration settings of glucose (50 mmol/L) and oleic acid (0.25 mmol/L) lack literature support or pre-experiment verification. It is necessary to supplement the explanation for the rationality of these concentration selections. For example, whether they refer to the common concentrations used in studies related to chicken hepatocyte nutritional metabolism, or whether the optimal treatment concentrations are determined through pre-experiments (such as investigating the effects of different concentration gradients on cell viability and triglyceride (TG) deposition) to avoid result deviations caused by inappropriate concentrations.
6. In the figure caption of Fig. 4, should be "Figure.4".
7. The Discussion section does not mention the limitations of the study. For instance, only 320-day-old recessive white-feathered chickens were selected for the in vivo experiment; and only chicken embryo hepatocytes were used in the in vitro experiment, which cannot fully simulate the physiological state of the liver in adult chickens. It is recommended to supplement the study limitations at the end of the Discussion.
8. The expression in the Conclusion section is overly repetitive. For example, "mitochondrial haplotypes affect the response of animals to changes in nutritional levels" is mentioned multiple times in the Conclusion. It is recommended to streamline the content of the Conclusion to enhance its conciseness and practicality.
9. The research conclusions regarding the two additives in the experiment have theoretical value. What practical guiding significance do they have for the formulation of animal feed? Please supplement the relevant content.
Reviewer 3 Report
Comments and Suggestions for Authors
Authors prepared an interesting study. However, number of hens 36 is extremely low. This is the main limitation of the study. It is not clear whether number of samples or replication in analysis is adequate. Authors must provide a power analysis to show that sample size is adequate to justify their experimental design and trials/analysis.
New references must be added.
Conclusion must be reduced by 50%. Its is not necessary to have repetition or such long conclusions. This part may be put in discussion part.
Reviewer 4 Report
Comments and Suggestions for Authors
In animal production, there is a considerable amount of research focused on nutrition, energy and mitochondria. Many related studies have been published. The research content of this article still has the following several issues:
- Nutritional fluctuations are usually regarded as the differences in nutritional components and energy in the feed. However, in this study, it refers to fasting and re-feeding. The term "nutritional fluctuations" in the title seems inappropriate. Consider revising and refining the content of the title.
- Each group in the experiment consisted of only 9 individuals, and the number of repetitions was too low. Animal experiments themselves have significant individual variations, and the small sample size affects the accuracy of the statistics.
- Is a fasting period of 10 or 12 hours sufficient? In poultry production, fasting usually requires a much longer duration.
- When conducting re-feeding, what is the feeding amount for each individual? Data on the total feeding amount should be provided.
- The conclusion should not merely be a summary of the results of each indicator, Instead, it should systematically condense the research content. Please further revise and streamline the conclusion of this article.
The language used in this article is overly verbose. The introduction and discussion sections are too lengthy. For instance, the second and third paragraphs of the introduction can be combined, and the fourth and fifth paragraphs can also be merged. Please revise the grammar and structure of this article to meet the publication requirements based on the writing guidelines for research papers.
Round 2
Reviewer 3 Report
Comments and Suggestions for Authors
Authors revised adequately.